# RICE: Refining Instance Masks in Cluttered Environments with Graph Neural Networks

**Christopher Xie**[1]   **Arsalan Mousavian**[2]   **Yu Xiang**[2]   **Dieter Fox**[1,2]
[1]University of Washington   [2]NVIDIA
{chrisxie,fox}@cs.washington.edu   {amousavian,yux}@nvidia.com

**Abstract:** Segmenting unseen object instances in cluttered environments is an important capability that robots need when functioning in unstructured environments. While previous methods have exhibited promising results, they still tend to provide incorrect results in highly cluttered scenes. We postulate that a network architecture that encodes relations between objects at a high-level can be beneficial. Thus, in this work, we propose a novel framework that refines the output of such methods by utilizing a graph-based representation of instance masks. We train deep networks capable of sampling informed perturbations to the segmentations, and a graph neural network, which can encode relations between objects, to evaluate the perturbed segmentations. Our proposed method is orthogonal to previous works and achieves state-of-the-art performance when combined with them. We demonstrate an application that uses uncertainty estimates generated by our method to guide a manipulator, leading to efficient understanding of cluttered scenes. Code, models, and video can be found at https://github.com/chrisdxie/rice.

**Keywords:** Unseen Object Instance Segmentation, Graph Neural Networks, Robot Perception

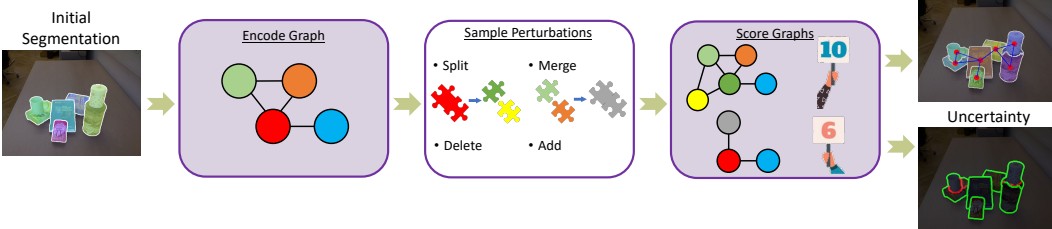

Figure 1: High-level overview of our proposed method. Given an initial segmentation, we encode it as a graph, sample perturbations, then score the resulting segmentation graphs. The highest scoring graph and/or contour uncertainty is output. Best viewed in color and zoomed in.

## 1 Introduction

Perception lies at the core of the ability of a robot to function in an unstructured environment. A critical component of such a perception system is its capability to solve Unseen Object Instance Segmentation (UOIS), as it is infeasible to assume all possible objects have been seen in a training phase. Proper segmentation of these unseen instances can lead a better understanding of the scene, which can then be exploited by algorithms such as manipulation [1, 2, 3] and re-arrangement [4].

Many methods for UOIS directly predict segments from raw sensory input such as RGB and/or depth images. While recent methods have shown strong results for this problem [5, 6, 7, 8], they still tend to fail when dealing with highly cluttered scenes, which are common in manipulation scenarios. A natural thought is that an architecture with relational reasoning can benefit the predictions. For example, it can potentially learn to recognize common object configurations (e.g. realizing that one object is stacked on top of another). While relational inductive biases have shown to be useful for problems such as scene graph prediction [9, 10, 11], it remains to be seen whether it can be useful in identifying objects in dense clutter. In this work, we investigate the use of graph neural networks, which can encode relations between objects, for segmenting densely cluttered unseen objects.

5th Conference on Robot Learning (CoRL 2021), London, UK.

In this paper, we propose a novel method for Refining Instance masks in Cluttered Environments, named RICE. Given an initial instance segmentation of unseen objects, we encode it into a *segmentation graph*, where individual masks are encoded as nodes and connected with edges when they are close in pixel space. Starting from this initial graph, we build a tree of sampled segmentation graphs by perturbing the leaves in a CEM-style (Cross Entropy Method) framework, where example perturbations include splitting and merging. We learn Sampling Operation Networks (SO-Nets) that sample efficient and informed perturbations that generally lead to better segmentations. The perturbed segmentation graphs are scored with a graph neural network, denoted Segmentation Graph Scoring Network (SGS-Net). Finally, we can return the highest scoring segmentation or compute contour uncertainties, depending on the application. Figure 1 provides a high-level illustration of our method.

RICE is able to improve the results of existing techniques to deliver state-of-the-art performance for UOIS. An investigatory analysis reveals that applying SGS-Net on top of the SO-Nets results in more accurate and consistent predictions. In particular, we find that SGS-Net learns to rank segmentation graphs better than SO-Nets alone. Additionally, we provide a proof-of-concept efficient scene understanding application that utilizes uncertainties output by RICE to guide a manipulator.

In summary, our main contributions are: 1) We propose a novel framework that utilizes a new graph-based representation of instance segmentation masks in cluttered scenes, where we learn deep networks capable of suggesting informed perturbations and scoring of the graphs. 2) Our method achieves state-of-the-art results for UOIS when combined with previous methods. 3) We demonstrate that uncertainty outputs from our method can be used to perform efficient scene understanding.

## 2   Related Work

**Instance Segmentation**   Traditional methods for 2D instance segmentation include GraphCuts [12], Connected Components [13], and LCCP [14]. Recently, learning-based approaches have provided more semantic solutions. For example, top-down solutions combine segmentation with object proposals in the form of bounding boxes [7, 15, 16, 17]. Mask R-CNN [7] predicts a foreground mask for each proposal produced by its region proposal network (RPN). However, when bounding boxes contain multiple objects (e.g. cluttered robot manipulation setups), the true instance mask is ambiguous and these methods struggle. Recently, a few methods have investigated bottom-up methods which assign pixels to object instances [18, 19, 20, 21, 22]. Some examples of this include contrastive losses [18] and unrolling mean shift clustering as a neural network to learn pixel embeddings [22].

Most of the afore-mentioned algorithms provide instance masks with category-level semantic labels, which do not generalize to unseen objects in novel categories. Class-agnostic methods [23, 24, 25, 26] and motion segmentation [27, 28, 29] methods have been investigated for this problem. In robotic perception, Xie et al. [30] proposed to separate the processing of depth and RGB in order to generalize their method from sim-to-real settings and provide sharp masks. Their follow-on work [5] proposed a 3D voting method to overcome the limitations of their earlier 2D method. Xiang et al. [6] showed that training a network on RGB-D with simulated data and a simple contrastive loss [18] can demonstrate strong results for this problem. While these methods show promise, they are not perfect and still admit mistakes in cluttered scenes, which can hamper downstream robot tasks that rely on such perception. Our method is orthogonal to these works, and is designed to refine their outputs by sampling perturbations to result in better instance segmentations in the cluttered environments.

**Graph Neural Networks**   Graph neural networks (GNN) in vision and robotics have recently become a useful tool for learning relational representations. They have found applications in many standard computer vision tasks such as image classification [31, 32], object detection [33], semantic segmentation [34], and question answering [35]. GNNs have also been used to perform "scene graph generation", which requires predicting not just object detections, but also the relations between the objects [9, 10, 11]. The resulting scene graphs have been used for applications such as image retrieval [36]. GNNs have also been used to learn object dynamics, properties, and relations for applications such as differential physics engines [37, 38]. Our proposed work represents instance segmentation masks as graphs and utilizes this architecture in order to refine the predicted masks.

## 3   Method

Our method, RICE, is designed to Refine Instance masks of unseen objects in Cluttered Environments. Given an initial segmentation mask $S \in \mathbb{N}^{H \times W}$ of unseen objects, we first encode this

as a *segmentation graph* $G_S$, which is described in Section 3.1. Then, in Section 3.2, we build a tree $T$ of sampled segmentation graphs by perturbing the leaves in a CEM-style [39] framework. Section 3.3 details the sampling operations, which are parameterized by our Sampling Operation Networks (SO-Nets). Each candidate graph (tree node) is scored by a GNN named Segmentation Graph Scoring Network (SGS-Net), introduced in Section 3.4. Finally, the highest scoring graph in $T$ and/or contour uncertainties are returned. Figure 1 provides a high-level illustration of RICE, and pseudocode can be found in the Supplement.

## 3.1 Node Encoder

Given a single instance mask $S_i \in \{0, 1\}^{H \times W}$ for instance $i$, we crop the RGB image $I \in \mathbb{R}^{H \times W \times 3}$, an organized point cloud $D \in \mathbb{R}^{H \times W \times 3}$ (computed by backprojecting a depth image with camera intrinsics), and the mask $S_i$ with some padding for context. We then resize the crops to $h \times w$ and feed these into a multi-stream encoder network which we denote as the Node Encoder. This network applies a separate convolutional neural network (CNN) to each input, and then fuses the flattened outputs to provide a feature vector $\mathbf{v}_i$ for this node. See Figure 2 for a visual illustration of the network. Note that we also encode the background mask as a node in the graph. This gives the segmenta-

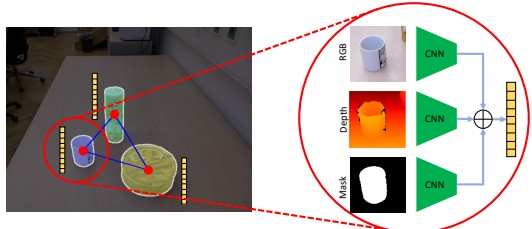

Figure 2: Given an initial instance segmentation mask (left), our segmentation graph representation encodes each individual mask as a graph node (red dots) with a corresponding feature vector $\mathbf{v}_i$ (yellow bar) output by the Node Encoder (right). Edges (blue lines) connect nearby masks.

tion graph $G_S = (V, E)$, where each $\mathbf{v}_i \in V$ corresponds to an individual instance mask, and nodes are connected with undirected edges $e = (i, j) \in E$ if their set distance is less than a threshold.

## 3.2 Building the Sample Tree

Our sample tree-building procedure operates in a CEM-style fashion. CEM [39] is an iterative sampling-based optimization algorithm that updates its sampling distribution based on an "elite set" of the top $k$ (or top percentile) samples. For more details, we refer the reader to [39]. Following this terminology, our elite set consists of the leaves of our sample tree $T$, each of which are guaranteed to be better with respect to our proxy objective function, SGS-Net. Then, the sampling distribution is implicitly defined by the SO-Nets; while we cannot explicitly write out the distribution, we can certainly sample from it with our sampling operations described in Section 3.3.

Our sample tree $T$ starts off with the root $G_S$. We expand the tree from the leaves with $K$ expansion iterations. For each expansion iteration, we iterate through the current leaves of $T$. For a leaf $G$, we randomly choose a sample operation from Section 3.3 and apply it to $G$ to obtain candidate graph $G'$. We then compare the scores $s_G, s_{G'}$ output by SGS-Net, and add $G'$ to $T$ as a child of $G$ if $s_{G'} > s_G$. Thus, any leaf of $T$ is guaranteed to be at least as good as the root $G_S$ w.r.t. our proxy objective function SGS-Net. We apply this procedure $B$ times for $G$, such that each tree node can have a maximum of $B$ children. Thus, $B$ is a branching factor. Finally, due to constraints of limited GPU memory, we exit the process in an anytime fashion whenever we exceed a budget of maximum graph nodes and/or graph edges (not to be confused with tree nodes/edges). See the Supplement for pseudocode and an example of the sample tree-building procedure.

It is important to note that while we utilize our learned SO-Nets and SGS-Net to build the sample tree $T$, they are applied in different manners (although they are trained on the same dataset). In order to add a candidate graph to the tree, they must both agree in the sense that the perturbation must be suggested via an SO-Net and SGS-Net must approve of the candidate graph via its score. This redundancy offers a level of robustness, Section 4.4 shows that the combination of these leads to more accurate performance with lower variance.

## 3.3 Sampling Operations

We consider four sampling operations: 1) splitting, 2) merging, 3) deleting, and 4) adding. However, randomly performing these operations leads to inefficient samples which wastes computation time and memory. For example, it is not clear how to split or add an instance mask randomly such that it may result in a better segmentation. Thus, we introduce two networks for these four operations, SplitNet and DeleteNet, which comprise our SO-Nets. They are learned to suggest informed perturbations to

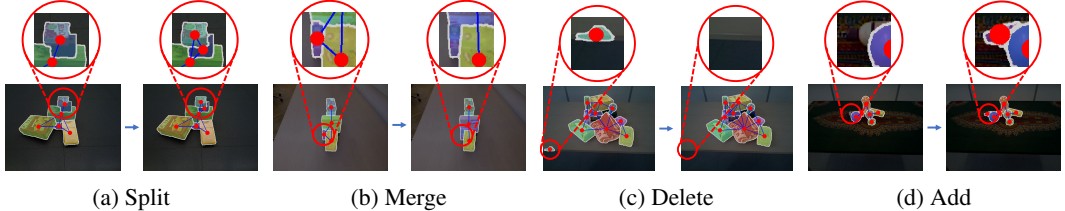

|  (a) Split | (b) Merge | (c) Delete | (d) Add |

Figure 3: We show real-world examples of the sampling operations and how they can refine the original segmentation. Best viewed in color on a computer screen and zoomed in.

bias the sampling towards better graphs, lowering the amount of samples needed in order to favorably refine the segmentation. Examples of each operation can be found in Figure 3.

**Split** It is not clear how to randomly split a mask such that it provides an effective split. For example, a naive thing to do is to sample a straight line to split the mask, however in many cases this will not result in a reasonable split (see Figure 3a for an example). Thus, we propose to learn a deep network denoted SplitNet to handle this. SplitNet takes the output of the Node Encoder (before flattening), fuses them with concatenation followed by a convolution, then passes them through a single decoder with skip connections. Essentially it is a multi-stream encoder-decoder U-Net [40] architecture, much like Y-Net [27], except that it has three streams for RGB, depth, and the mask. The output of SplitNet is a pixel-dense probability map $p_i \in [0, 1]^{h \times w}$ of split-able object boundaries. To sample a split for instance mask $S_i$, we first sample two end points on the contour of the original mask $S_i$, and calculate the highest probability path from the end points that travels through $p_i$, resulting in a trajectory $\tau = \{(u_t, v_t)\}_{t=1}^{L_i}$ of length $L_i$. We score the split with $s_\tau = \frac{1}{L_i} \sum_t p_i[u_t, v_t] \in \mathbb{R}$, which is the average probability along the sampled path. More details can be found in the Supplement.

**Merge** We exploit the fact that merging is the opposite of splitting and adapt SplitNet for this operation. For each pair $(i, j)$ of neighboring masks, we take their union $S_{ij}$ and pass it through SplitNet to get $p_{ij}$. Note that we do not consider merging disjoint masks that may belong to the same instance, which is a limitation of this work. To compute the merge score $m_{ij}$, we first compute the union of the boundaries of $S_i$ and $S_j$, denoted $B_{ij} \in \{0, 1\}^{h \times w}$. Then, we calculate the merge score as $m_{ij} = 1 - (p_{ij} \odot B_{ij}/(\mathbf{1}^\mathsf{T} p_{ij} \mathbf{1}))$ where $\odot$ is element-wise multiplication, $\mathbf{1}$ is a vector of ones. This is essentially a weighted average of $B_{ij}$ with weights $p_{ij}$. This score indicates how likely SplitNet thinks $S_i$ and $S_j$ correspond to different objects. Figure 3b shows an ideal merge operation.

**Delete** We design a network, DeleteNet, to provide delete scores $d_i \in \mathbb{R}$ for every instance (graph node) $i$. This network is also built on top of the Node Encoder: it computes the difference $\mathbf{v}_i - \mathbf{v}_{bg}$, where $\mathbf{v}_{bg}$ is the feature vector for the background node output by the Node Encoder. This difference is then provided as input to a multi-layer perceptron (MLP) which outputs a scalar $d_i$. See Figure 3c for an example of how DeleteNet can help remove false positives from the segmentation.

**Add** Similarly to merging, we can exploit the fact that adding is the opposite of deleting. Given a candidate mask $S_{N+1}$ to add to the graph, we can use DeleteNet to compute its delete score $d_{N+1}$. If $d_{N+1}$ is below a threshold, we successfully add the mask to the graph. However, the question remains of how to generate such candidate masks. Given an external foreground mask $F \in \{0, 1\}^{H \times W}$ (provided by UOIS-Net-3D [5]), we run connected components on $F \setminus \{\cup_i S_i\}$, and use the discovered components as potential new masks. A successful addition operation can be seen in Figure 3d.

### 3.4 Segmentation Graph Scoring Network

While our sample operations provide efficient samples that typically lead to better segmentation graphs, they can also suggest samples that worsen the segmentation. Thus, we learn SGS-Net which acts as a proxy for the objective function in the CEM framework. Our proposed SGS-Net learns to score a segmentation graph by considering the fused feature vectors $\mathbf{v}_i$ in context of their neighboring graph nodes (masks). We posit that this context will aid SGS-Net in predicting whether the perturbations improve the segmentation. For example, it can potentially learn to recognize common object configurations from the training set, and score such configurations higher.

A high-level illustration of SGS-Net can be found in Figure 4. The initial node features $\mathbf{v}_i^{(0)}$ are given by the Node Encoder, and we obtain initial edge features $\mathbf{e}_{ij}^{(0)}$ by running the Node Encoder on

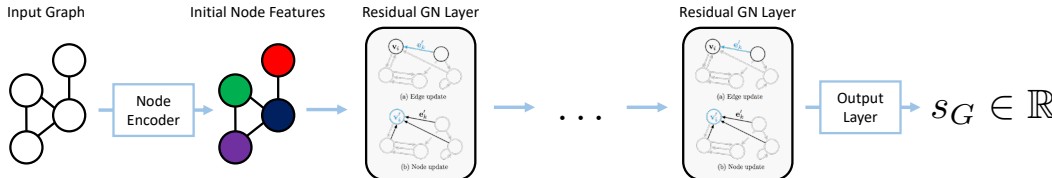

Figure 4: A high-level illustration of our Segmentation Graph Scoring Network (SGS-Net). It is composed of a Node Encoder (see Figure 2), multiple Residual GraphNet Layers, and an output layer. We borrowed elements from Figure 3 of Battaglia et al. [41].

all neighboring union masks $S_{ij}$. Then, we pass them through multiple Residual GraphNet Layers (RGLs), which are essentially GraphNet Layers [41] with a residual connection. We refer readers to Battaglia et al. [41] for details of GraphNet Layers, and also provide a full mathematical specification of RGLs in the Supplement for completeness. The output of SGS-Net is a scalar score in $[0, 1]$.

### 3.5 Training Procedure

For SplitNet, we apply a weighted binary cross entropy (BCE) loss to the probability map $p$: $\ell_{\text{split}} = \sum_u w_u \, \ell_{bce}\left(p_u, \hat{p}_u\right)$ where $u$ ranges over pixels, $\hat{p} \in \{0, 1\}^{h \times w}$ is ground truth boundary, and $\ell_{bce}$ is the binary cross entropy loss. The weight $w_u$ is inversely proportional to the number of pixels with labels equal to $\hat{p}_u$. DeleteNet is also trained with standard BCE loss. SGS-Net is trained with $\ell_{bce}$ to regress to $.8F + .2F@.75$, where $F$ is the Overlap F-measure [30] and $F@.75$ is the Overlap $F@.75$ measure [42]. The latter measures the percentage of correctly segmented instances. Thus, SGS-Net learns to predict a score based on the number of correctly identified pixels and instances. Note that this regression problem is very difficult to solve. However, the scores do not actually matter as long as the relative scoring is correct, since building the sample tree relies only on this (Section 3.2). In Section 4.5 we show that while SGS-Net may not solve the regression problem well, it learns to rank graphs accurately. Further training and implementation details can be found in the Supplement.

## 4 Experiments

### 4.1 Encoding RGB and Modality Tuning

We use ResNet50 [43] with Feature Pyramid Networks [44] (FPN) to encode RGB images before passing them to the Node Encoder. However, since we are training with (a more cluttered version of) the non-photorealistic synthetic dataset from Xie et al. [30], we perform modality tuning [45], where we fine-tune earlier convolutional layers of ResNet50 during training, and use the COCO [46] pretrained weights during inference. For all experiments, we modality tune the `conv1` and `conv2_1` blocks of ResNet. We provide an experiment in the Supplement that shows this setting is optimal.

### 4.2 Datasets and Metrics

We evaluate our method on two real-world datasets of challenging cluttered tabletop scenes: OCID [47] and OSD [48], which have 2346 images of semi-automatically constructed labels and 111 manually labeled images, respectively. Our SO-Nets and SGS-Net are trained on a more cluttered version of the synthetic Tabletop Object Dataset (TOD) [30], where each scene has anywhere between 20 and 30 ShapeNet [49] objects. We use 20k scenes in total, with 5 images per scene.

Xie et al. [30] introduced the Overlap P/R/F and Boundary P/R/F measures for the problem of UOIS. However, these metrics do not weight objects equally; they are dependent on the size and larger objects tend to dominate the metrics. Thus, we introduce a variation to these metrics that equally weights the errors of individual objects regardless of their size. Given a Hungarian assignment $A$ between the predicted instance masks $\{S_i\}_{i=1}^N$ and the ground truth instance masks $\{\hat{S}_j\}_{j=1}^M$, we compute our Object Size Normalized (OSN) P/R/F measures as follows:

$$P_n = \frac{\sum\limits_{(i,j) \in A} P_{ij}}{N}, \quad R_n = \frac{\sum\limits_{(i,j) \in A} R_{ij}}{M}, \quad F_n = \frac{\sum\limits_{(i,j) \in A} F_{ij}}{\max(M, N)}, \quad F_n@.75 = \frac{\sum\limits_{(i,j) \in A} \mathbf{1}\{F_{ij} >= 0.75\}}{\max(M, N)},$$

where $P_{ij}, R_{ij}, F_{ij}$ are the precision, recall, and F-measure of $S_i, \hat{S}_j$. Note that the $F_n@.75$ penalizes both false positive and false negative instances, as opposed to the normal $F@.75$, which does not penalize false positives. Similarly to Xie et al. [30], we can apply the OSN metrics to the pixels and

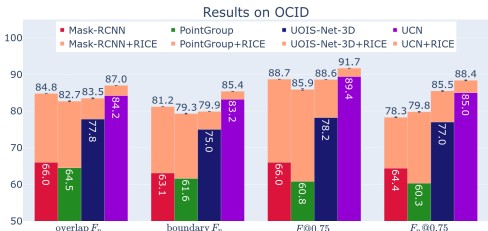 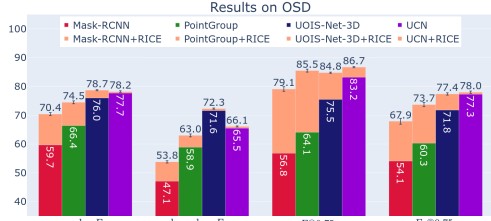

Figure 5: Applying RICE to refine results from state-of-the-art instance segmentation methods leads to improved performance across the board. Note that standard deviation bars are shown, but are very tight and difficult to see.

| SO-Nets | SGS-Net | Overlap | | | Boundary | | | $F@0.75$ | $F_n@0.75$ |
|---------|---------|---------|---------|---------|----------|---------|---------|----------|-----------|
| | | $P_n$ | $R_n$ | $F_n$ | $P_n$ | $R_n$ | $F_n$ | | |
| ✗ | ✗ | 85.1 (–) | 83.0 (–) | 77.8 (–) | 84.6 (–) | 76.5 (–) | 75.0 (–) | 78.2 (–) | 77.0 (–) |
| ✓ | ✗ | 84.7 (1.23) | 89.4 (0.19) | 82.3 (1.09) | 82.7 (1.37) | 82.8 (0.19) | 78.7 (1.10) | 89.0 (0.26) | 84.2 (1.26) |
| ✓ | ✓ | 86.3 (0.03) | 89.1 (0.01) | 83.6 (0.05) | 84.5 (0.04) | 82.5 (0.04) | 80.0 (0.04) | 88.5 (0.02) | 85.5 (0.05) |

Table 1: Ablation to test the utility of SO-Nets and SGS-Net on OCID [47] starting from UOIS-Net-3D [5] masks. Only using the sample operator networks (SO-Nets) in an iterative sampling scheme already provides an increase in performance, showing that the informed samples are generally improving the initial segmentations. However, the standard deviations (shown in parentheses) are relatively high. Adding in SGS-Net boosts performance while drastically lowering the variance, demonstrating the efficacy of SGS-Net in consistently filtering out bad suggestions by the SO-Nets.

boundaries, giving us Overlap and Boundary $P_n/R_n/F_n$ measures. For comparison, we also show results with the normal Overlap and Boundary P/R/F measures in the appendix.

We run each experiment 5 times and show means and standard deviations for all metrics.

### 4.3 SOTA Improvements

We demonstrate how RICE can improve upon predicted instance segmentations from state-of-the-art methods. In particular, we apply it to the results of Mask R-CNN [7], PointGroup [8], UOIS-Net-3D [5], and UCN [6], all of which consume RGB-D as input. We employ RICE by returning the best segmentation. For brevity, we only show Overlap $F_n$, Boundary $F_n$, $F@.75$, and $F_n@.75$ in Figure 5 on both OCID and OSD. The light orange bars show the additional performance that RICE provides over the output of the methods. Standard deviations are shown as error bars, but are in general very narrow, showing that our method provides consistent results despite its stochasticity. RICE provides substantial improvements to all methods. The largest gains occur in Mask R-CNN and PointGroup, with 21.6% and 32.3% relative gain in $F_n@.75$ on OCID, respectively. Additionally, on the already strong results from UOIS-Net-3D and UCN, RICE achieves 11.0% and 4.0% relative gain in $F_n@.75$ on OCID, respectively. These results are similar on OSD, with the gains being slightly less pronounced, which we believe is due to OSD being a smaller dataset with less clutter. Note that applying RICE increases both $F@.75$ and $F_n@.75$, indicating that not only is it capturing the object identities correctly, it is not simultaneously predicting more instances (false positives). In the appendix, we show full results for all metrics including $P_n, R_n$, and normal P/R/F metrics.

### 4.4 Ablation Study

We aim to answer two questions with this study: 1) how good are the samples suggested by our SO-Nets, and 2) to what degree does SGS-Net increase performance and robustness? We study these questions on the larger OCID.

Since the SO-Nets alone do not provide scores of the perturbed segmentation graphs, we structure our ablation such that this is not needed in order to answer 1). Our SO-Nets are trained to provide informed perturbations that are closer to the ground truth segmentation, so every sample is supposed to be better than the original graph. With this insight, we design an experiment where we run RICE with branch factor $B = 1$ and $K = 5$ iterations, always add the candidate graph to the tree without consulting SGS-Net, and return the final graph. Essentially, this can be seen as an iterative segmentation graph refinement procedure where the sampled graph should be better than the previous in every iteration. Starting from initial masks provided by UOIS-Net-3D [5], we see in Table 1 that applying this iterative sampling scheme with SO-Nets only provides better results on almost all

metrics than without. However, adding SGS-Net back into the procedure results in better Overlap $F_n$, Boundary $F_n$, and $F_n$@.75, while significantly reducing the standard deviation of the results by two orders of magnitude. This demonstrates that having SGS-Net in RICE delivers not only more accurate performance, but also more robust performance with relatively small variance, which answers 2). Note that $F$@.75 is slightly lower with $F_n$@.75 higher, indicating that SO-Nets are suggesting more samples that better capture the objects, but are suggesting too many instance segments.

### 4.5 SGS-Net Ranking

Figure 6 shows an example of how difficult scoring the segmentations graphs is; the two slightly different segmentations have a significant difference in their ground truth scores. In fact, SGS-Net does a poor job at scoring the graphs, with a mean absolute error (MAE) of 0.184 and even higher standard deviation shown in Table 2. These values are high given that the scores are in the range $[0, 1]$. Then, this begs the question, why does SGS-Net work well within our proposed RICE framework? Recall that

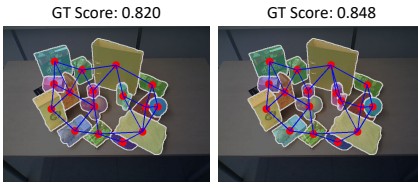

Figure 6: Can you spot the differences between the segmentations?

the score magnitudes do not matter, only the relative scoring (Section 3.2). We claim that SGS-Net learns to rank the graphs accurately, and design an experiment to test this hypothesis.

We leverage the normalized Discounted Cumulative Gain (nDCG) [50] which is a popular ranking metric in the information retrieval community. The DCG is computed as $\sum_{i=1}^{p} \frac{2^{\text{rel}_i} - 1}{\log_2(i+1)}$ where $\text{rel}_i$ is the numerical relevance (1 to $p$, higher is better) of the item at position $i$. This essentially computes a weighted sum of the relevance with a discount

|  | MAE | nDCG |
|---|---|---|
| Minimum | – | 0.844 (0.196) |
| SO-Nets | – | 0.944 (0.098) |
| SGS-Net | 0.184 (0.212) | 0.952 (0.095) |

Table 2: Ranking study on OCID and OSD.

factor for further items, which places more emphasis on the high-ranking predictions. The normalized version divides DCG by the "ideal" version, i.e. the DCG of the correct ranking. This results in nDCG $\in [0, 1]$ with higher being better. We compute nDCG of the ranking of the iterative sampling experiment in Section 4.4, with relevance values in $\{0, ..., K\}$. The ranking for SO-Nets is given by the order of the predicted graphs, and we use SGS-Net scores to compute its ranking/relevance. We also compute the nDCG of the worst ranking, denoted "minimum". In Table 2, we see that both SO-Nets and SGS-Net perform significantly better than the worst ranking. SGS-Net provides better ranking than SO-Nets with slightly lower variance, which helps to explain its effectiveness in RICE.

### 4.6 Visualizing Refinements

In the left side of Figure 7 (green box), we qualitatively demonstrate successful refinements from applying RICE to instance masks provided by state-of-the-art methods. The first column shows an example where many nearby objects are under-segmented. Indeed, RICE manages to find all of the necessary splits except for one. In general, RICE is quite adept at splitting under-segmented instance masks. This is quantitatively confirmed in an additional ablation in the Supplement that studies the usefulness of each sampling operation. Column two shows an initial mask that is fixed with a merge operation. Column three shows a false positive mask on the textured background, which is suppressed by RICE's deletion sampling operation. In the fourth column, the initial mask is missing quite a few objects, and RICE is able to not only recover them but also correctly segment them, resulting in an almost perfect instance segmentation. In the last column, the bottom left segment is bleeding into a neighboring segment, which is fixed through multiple perturbations (i.e. split, then merge).

### 4.7 Failures and Limitations

In the right side of Figure 7 (red box), we discuss some failure modes and limitations. The first column demonstrates a failure mode where RICE tends to over-segment objects with a lot of texture (e.g. cereal box). We believe that this is due to TOD lacking texture on many of its objects [5]. The second column shows a limitation: since RICE only considers merging neighboring masks, it cannot merge non-neighboring masks that belong to the same object. RICE does nothing and the book is still incorrectly segmented in two pieces. We leave this as an interesting avenue for future work.

### 4.8 Guiding a Manipulator with Contour Uncertainties for Efficient Scene Understanding

Fully segmenting and understanding a scene of cluttered objects is necessary for various manipulation tasks, such as counting objects or re-arranging and sorting them. One way for doing this is to actively

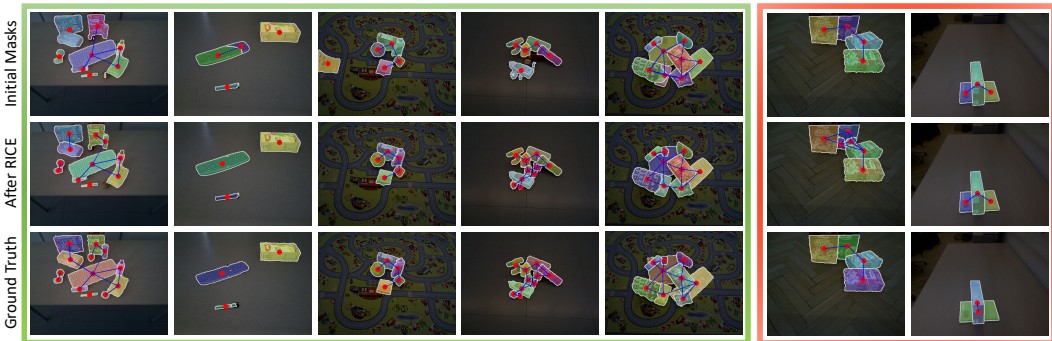

Figure 7: We demonstrate successful refinements (left, green box) for each of the sampling operations. Failure modes (right, red box) include textured objects and non-neighboring masks that belong to the same object. Best viewed in color and zoomed in on a computer screen.

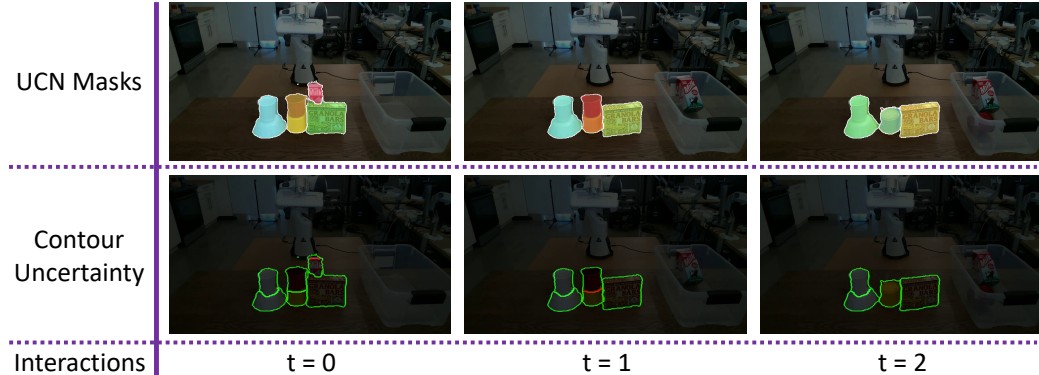

Figure 8: UCN masks [6] (top row) and contour uncertainties from RICE (bottom row, uncertainties are shown in red with average contours in green) in a trial of our scene understanding experiment. After grasping the milk carton and red cup, the scene is segmented with full certainty, indicating that the scene is fully understood. Thus, the algorithm terminates without having to singulate each object.

singulate each object [51]. However, such an approach can be extremely inefficient. Here we show how contour uncertainties extracted from RICE can help to solve this problem with potentially far less interactions. Specifically, we extract contour uncertainties by computing the standard deviation of the mask contours of each leaf graph. These uncertainties let us distinguish between objects that are already confidently segmented and those that require physical interaction to resolve segmentation uncertainty. We grasp [52] any object that has uncertain contours in order to determine its correct segmentation, and repeat this until no more uncertainty persists. Thus, interactions are only required to resolve the uncertain portions of the scene, which can potentially be much less than the number of objects, leading to a more efficient scene understanding method. For example, in Figure 8, only two grasps are required to fully understand the scene. See the Supplemental video for more results.

## 5  Conclusion and Future Work

We have proposed a novel framework that utilizes a graph-based representation of instance segmentation masks. It incorporates deep networks capable of sampling smart perturbations, and a graph neural network that exploits relational inductive biases. Our experimental analysis revealed insight into why our method achieves state-of-the-art performance when combined with previous methods. We further demonstrated that our uncertainty outputs can be utilized to perform efficient scene understanding.

A main limitation of our work is the computational burden; the algorithm runs at 10-15 seconds per frame, depending on the expansion of the sample tree. Additionally, it is GPU-memory intensive as the sample tree must be stored in GPU memory. Future work will explore how to make the method more computationally efficient, along with solving the inherent limitations mentioned in Section 4.7.

## Acknowledgements

This work was funded in part by NSF NRI grant IIS-2024057.

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
