# OpenReview forum: "RICE: Refining Instance Masks in Cluttered Environments with Graph Neural Networks"
_robot-learning.org/CoRL/2021/Conference — CoRL2021 Poster_

### Official Review · Reviewer_cMBw · 2021-07-18

**Originality:** Good
**Technical Quality:** Good
**Clarity Of Presentation:** Very Good
**Impact:** 1

**Recommendation:**

Weak Accept: I recommend accepting the paper, but will not argue for my recommendation if the majority of other reviewers have a different opinion.

**Summary:**

This paper proposes an approach for image segmentation using graph-based representation and several deep neural networks trained to perturb and score different image segmentations. I like the framework architecture in general, and the results look very promising. But I'm a bit suspicious that this paper is more suited for a computer vision venue than for a robotics one. However, I'm going to give the authors the benefit of the doubt and vote for acceptance.

**Issues:**

- In Line 98, h and w are not defined.

**Reviewer Expertise:**

Fair: Some knowledge of the area

**Strengths And Weaknesses:**

Strengths:
- The approach seems straightforward to re-produce.
- The paper is well-written and organized.

Weaknesses:
- As mentioned by the authors, the framework is pretty computation-heavy at the moment.
- The robotics experiment is very simplistic and it uses an out-of-the-box grasping method (probably developed by the authors in their previous work).

**Summary Of Recommendation:**

The robotic experiments seem very simplistic to me. But the segmentation approach is very interesting and the results look good, and since this can have lots of applications in robotics, I vote for acceptance.

---

> ### Author Response · Authors · 2021-08-26
> **Response to Reviewer cMBw**
>
> We thank the reviewer for the comments.
>
> > I'm a bit suspicious that this paper is more suited for a computer vision venue than for a robotics one.
>
> While the proposed framework uses vision and ML techniques to improve on the problem of unseen object instance segmentation, the method is directly motivated by robot requirements. The settings we operate in are meant for manipulation, not in-the-wild computer vision. Our method exploits the structure of cluttered manipulable objects (via GNNs), which are common to robot manipulation environments, and not necessarily in in-the-wild computer vision settings. Thus, our results are directly applicable to robots, and we believe this work is more relevant and suitable for a robotics venue.
>
> > As mentioned by the authors, the framework is pretty computation-heavy at the moment.
>
> Bringing this framework closer to real-time is an avenue of further research.
>
> > The robotics experiment is very simplistic and it uses an out-of-the-box grasping method.
>
> The main contribution of this work is the introduction of the framework for improving instance segmentations. Thus, most of the experimental section focuses on that. The robot experiment is a proof of concept that shows how the uncertainty estimates can be used to efficiently singulate and understand objects. There are many potential robotics applications that can build off both the instance segmentation (e.g. [1,2,3,4]) and the uncertainty estimates, and we hope that the community will build off of our advances to solve such tasks.
>
> > In Line 98, h and w are not defined.
>
> In L54 in the Supplement, we detail the implementation details of this. h and w are set to 64.

---

### Official Review · Reviewer_1fhJ · 2021-07-19

**Originality:** Very Good
**Technical Quality:** Excellent
**Clarity Of Presentation:** Very Good
**Impact:** 4

**Recommendation:**

Strong Accept: I recommend accepting the paper and will argue for my recommendation even if other reviewers hold a different opinion.

**Summary:**

The authors propose a GNN-based refinement method for unknown instance segmentation based on RGB-D data. Therefore, they build a graph by connecting segments that are close in pixel space. Then they perform an informed perturbation of that graph by splitting, merging, adding and/or deleting segments and finally rank the resulting graphs using another GNN. It is shown that the method can successfully improve results of recent unknown instance segmentation methods. The approach is applied to object singulation where it helps a robot to only singulate the objects with uncertain contours.

**Issues:**

- "smart perturbations" are used throughout the paper. I would rather say "informed perturbations" or "learned perturbations".
- l275 "numerical relevance" should be defined as it is otherwise unclear and hard to find in the reference
- Table 2: Is the worst ranking really a good comparison?
- Figure 7: It is annoying to scroll up and down to get the description of the figure, which is actually in section 4.6. It would be great if you could move the descriptive part in the caption.

Since code release was not announced:
- l97: padding is not specified
- l108: threshold is not specified


**Reviewer Expertise:**

Excellent: Expert knowledge on the topic of the paper

**Strengths And Weaknesses:**

Strengths:

- A refinement of unknown segmentation masks by taking the whole scene into account had been missing in the field and the GNN-based approach is an expensive but convincing attempt.
- The details how the network learns to predict splits, merges, adds and deletes are interesting
- The ablations (also in the detailed appendix) show the impact of individual components
- The method improves upon different baseline methods in the RGB-D domain on OCID and OSD
- Previously used performance metrics are corrected to be independent to object size.
- The approach is demonstrated on a robotic system
- The paper is clearly written and seems relatively honest wrt. limitations

Weaknesses (of the approach, paper issues will be listed below):

- The large number of networks and forward passes due to the sampling operations make this method quite slow and memory-intensive to use in robotic applications (10-15s/frame). However, the authors and readers are aware of that and it is a good proof of concept.
- The networks are completely agnostic to the approach that generated the initial masks. On the one hand, this is beneficial because the refinement can be easily used for different initial methods. On the other hand, it leads to a lot of redundancy, as you need a new feature extractor that learns what an object is. Also using the feature representation of UCN probably contains richer information than just a binary mask and the representation could be directly refined end-to-end by the GNN scores.
- In the robotic experiments it seems like the network is able to merge/split/delete existing masks but has more troubles in adding completely new segments or completing undersegmented masks by adding+merging. This limitation could be made more clear, especially by showing separate results for adding and deleting in Table 1 of the appendix. The problem is probably the generation of candidate masks. I would be curious if an oversegmentation through low thresholds in the initial methods and subsequently only merging/splitting/deleting wouldn't lead to better results.
- The approach is trained on non-photorealistic data (perhaps for better comparison) and then needs to be manually adapted to work better with real data. There might be a more elegant solution, maybe by directly training on more realistic data.
- It would be interesting how the method deals with many similar objects

**Summary Of Recommendation:**

If my issues are resolved I recommend a strong accept. The paper presents an original idea on a relevant problem which has not been studied enough. The experiments are thorough and it is shown how the approach could benefit a robotic system.

---

> ### Author Response · Authors · 2021-08-26
> **Response to Reviewer 1fhJ**
>
> We thank the reviewer for the detailed, helpful comments.
>
> > The large number of networks and forward passes due to the sampling operations make this method quite slow and memory-intensive to use in robotic applications (10-15s/frame). However, the authors and readers are aware of that and it is a good proof of concept.
>
> Bringing this framework closer to real-time is an avenue of further research.
>
> > The networks are completely agnostic to the approach that generated the initial masks. On the one hand, this is beneficial because the refinement can be easily used for different initial methods. On the other hand, it leads to a lot of redundancy, as you need a new feature extractor that learns what an object is. Also using the feature representation of UCN probably contains richer information than just a binary mask and the representation could be directly refined end-to-end by the GNN scores.
>
> This is a great point. Using a feature extractor based on UCN could potentially give more information compared to a binary mask, especially because UCN was trained in tabletop settings as well. However, our pre-trained feature extractor is ResNet50+FPN trained on COCO, which provides RICE with features trained on a photorealistic dataset. UCN is trained on non-photorealistic synthetic data; it works quite well in the real world (when consuming RGB-D), but it is unclear how well the RGB-only features will transfer to real-world settings. Combining both feature representations could be interesting, but even more computationally expensive. Exploring this is an interesting avenue of future work.
>
> > In the robotic experiments it seems like the network is able to merge/split/delete existing masks but has more troubles in adding completely new segments or completing undersegmented masks by adding+merging. This limitation could be made more clear, especially by showing separate results for adding and deleting in Table 1 of the appendix. The problem is probably the generation of candidate masks. I would be curious if an oversegmentation through low thresholds in the initial methods and subsequently only merging/splitting/deleting wouldn't lead to better results.
>
> While this is true, it is also the case that the need to complete undersegmented masks (even adding masks) is relatively rare given the initial segmentation methods (Mask R-CNN, PointGroup, UOIS-Net, UCN). This is briefly discussed in L98-99 of the Supplement. Columns 4 and 5 of Figure 7 (main paper) show examples where combinations such as adding+splitting and splitting+merging are used to fix the scene. Using an oversegmentation with RICE is an interesting exploration of future work, although this would generate many nodes and potentially cause the GPU to run out of memory. We will make this more clear in the Supplement.
>
> > The approach is trained on non-photorealistic data (perhaps for better comparison) and then needs to be manually adapted to work better with real data. There might be a more elegant solution, maybe by directly training on more realistic data.
>
> We completely agree. However, obtaining annotated data for specific settings such as tabletop settings is currently resource intensive and costly which is not possible for academic labs.
>
> > "smart perturbations" are used throughout the paper. I would rather say "informed perturbations" or "learned perturbations".
>
> Thank you for the better word choice. We will make this change in a revision.
>
> > L275 "numerical relevance" should be defined as it is otherwise unclear and hard to find in the reference
>
> “Numerical relevance” is essentially a ranking (e.g. 1 to 5, with 5 being the best). We will clarify this in a revision.
>
> > Table 2: Is the worst ranking really a good comparison?
>
> It simply gives a base of which to ground the nDCG numbers to. Essentially, it says that 0.844 is the worst while 1.0 is the best. Then the numbers in Table 2 can be better understood.
>
> > Figure 7: It is annoying to scroll up and down to get the description of the figure, which is actually in section 4.6. It would be great if you could move the descriptive part in the caption.
>
> Thank you for pointing this out. We will fix this in a revision.
>
> > Since code release was not announced: L97: padding is not specified. L108: threshold is not specified.
>
> Our code/data/models (including code for training the models) are prepped and ready for release. We plan on releasing the code when the paper is accepted for publication. These details can be found in the code.

---

> > ### Comment · Reviewer_1fhJ · 2021-08-30
> > **Thank you for the clarifications**
> >
> > My only remaining question is really how the deleting and adding operations individually influence the performance. I understand that they are conceptually very similar, but it would still be interesting for the reader.

---

> > > ### Author Response · Authors · 2021-08-30
> > > **Response to Reviewer 1fhJ**
> > >
> > > Ahh, thank you for clarifying your question. We will run the experiment and edit Table 1 in the Supplement accordingly.

---

### Official Review · Reviewer_5yWX · 2021-07-23

**Originality:** Good
**Technical Quality:** Good
**Clarity Of Presentation:** Very Good
**Impact:** 3

**Recommendation:**

Weak Accept: I recommend accepting the paper, but will not argue for my recommendation if the majority of other reviewers have a different opinion.

**Summary:**

In this paper, the authors propose a neural-network-based method to refine segmentation masks provided by different models. In particular, their goal consists in improving unknown object segmentation masks in cluttered environments, where classical methods fail.The idea consists in (1) representing masks as graphs and use a GNN that produces variations of input masks, and (2) use an additional DNN to evaluate such variations and select them. The set of variations is generated iteratively through sampling and using CEM.

**Issues:**

* Section 3.2 is difficult to understand as it is currently placed. It would be better placed after 3.4
* Why not using IOU as a metric? For example, https://arxiv.org/pdf/2103.06796.pdf uses it.
* The paper does not report how many training runs have been performed: if results are average results, then std-dev is lacking
* How do the hyper-parameters affect the performance?

**Reviewer Expertise:**

Good: General knowledge of the area

**Strengths And Weaknesses:**

PROS:
* The paper is well structured and well written
* The idea is interesting and easy to apply, as it does not have particular requirements
* The proposed method can be applied (post-training) to several architectures to improve their results
* Although the approach is stochastic, the sampling statistic is is low and predictions are stable
* Experimental evaluation on 4 architectures, 2 of which are recent and task-optimized
* Experimental evaluation includes a use-case showing the utility of the method

CONS:
* std-dev is studied in the sampling procedure, but it is unclear on how many trial it is computed
* The paper does not report how many training runs have been performed: if results are average results, then std-dev is lacking
* The approach makes use of some hyper-parameters (e.g., branching factor, vicinity threshold, max node budget): there is no analysis on how those affect the results




**Summary Of Recommendation:**

The paper is well written and well structured. The proposed approach is interesting, easily applicable, and well supported by
experimental evaluations. This said, the document could be further improved by performing in-depth analysis of hyper-parameters
as well as better studying the average/std-dev of the training performance.

---

> ### Author Response · Authors · 2021-08-26
> **Response to Reviewer 5yWX**
>
> We thank the reviewer for the helpful notes.
>
> > std-dev is studied in the sampling procedure, but it is unclear on how many trial it is computed
>
> As mentioned in L223, each experiment is run 5 times. Averages and standard deviations are reported from these trials.
>
> > Section 3.2 is difficult to understand as it is currently placed. It would be better placed after 3.4
>
> We are open to re-arrange accordingly if the consensus is that reordering helps making the paper easier to follow. Additionally, we provide pseudocode for the entire framework in Algorithm 2 in the Supplement to supply the reader with a clearer understanding of RICE.
>
> > Why not using IOU as a metric?
>
> IoU is a valid metric for this task. However, it is similar (highly correlated) to the overlap F measure reported in Table 1 and Figure 5, thus it is somewhat redundant and we decided not to report it. Calculating IoU would require a Hungarian matching of predicted and ground truth masks, which the overlap F measure also computes.
>
> > The paper does not report how many training runs have been performed
>
> We did not report multiple training runs. This is standard in the deep learning literature. While we are using a stochastic optimization algorithm (Adam) to optimize our networks, different training runs with random initializations lead to very similar results.
>
> > How do the hyper-parameters affect the performance?
>
> We studied the effect of the major design choices such as the inclusion of SO-Nets and SGS-Net, modality tuning, and the sampling operations. Most other hyperparameters do not affect performance, but instead are chosen due to limitations in GPU memory. For example, branching factors and max node budgets are chosen to prevent the method from running into Out-Of-Memory errors on the GPU.

---

### Official Review · Reviewer_QM46 · 2021-07-25

**Originality:** Good
**Technical Quality:** Fair
**Clarity Of Presentation:** Very Good
**Impact:** 3

**Recommendation:**

Weak Reject: I recommend rejecting the paper, but will not argue for my recommendation if the majority of other reviewers have a different opinion.

**Summary:**

The paper presents a refinement method for instance masks in cluttered environments. The method uses the relationship between overlapping objects in a cluttered scene to improve the instance segmentation of each individual object. The presented algorithm combines a graph neural network and combinatorial optimization to find an ideal set of interacting instance masks.

**Issues:**

See issues above.

In addition, it would be nice to have a more detailed comparison of SplitNet and the entire pipeline, as the performance if split alone is almost the same as the complete combinatorial optimization.

**Reviewer Expertise:**

Good: General knowledge of the area

**Strengths And Weaknesses:**

+ Well written
+ Good improvement over multiple baseline algorithms (general)
- Unlikely reproducible
- Somewhat arbitrary

The paper is generally well written and easy to follow. The presented algorithm is interesting, and yields a good improvement over multiple baselines.

While the paper does a good job explaining things that are contained in it, it lacks some details that would make both experiments and method reproducible. Are the authors open to releasing code and data upon a potential acceptance?
Some of the details that were unclear from the current submission: What data-augmentation is used when training the different SO-Nets?
 * How are 'training' samples constructed? Do they take into account potentially wrong initial segments of baseline detectors?
 * How many of the baselines use the same input modalities (RGB-D) as the presented method? How many use just RGB or D? Might some of the gains come from these additional modalities?
 * Could RICE work from scratch? Why does it need a baseline detector?
Even with the additional details in the supplement, it would be hard to reproduce without code, which greatly diminishes the value of the submission.

Finally, the method is at times a bit arbitrary. There are many design choices that are simply states, and neither motivated or ablated. For example:
 * SplitNet could as well just predict a binary mask instead of its boundary that needs to be sampled and optimized. I understand that there is some ambiguity in the prediction (both the binary mask and its complement are valid answers), but this can easily be broken using either a single positive example, a hand-coded ordering (left is mask 1, right is mask 2), or even a Hungarian loss. It would be nice to know that the presented boundary prediction is in deed somewhat optimal. This is especially important since most of the presented method replies in a good split (and most other components either hurt or do not significantly improve the performance, see supplement).
* In a similar spirit, the AddNet relies on a proposal constructed from a connected component, why not use an RPN?


**Summary Of Recommendation:**

The paper starts with an interesting idea, but does not provide sufficient evidence to show the efficacy of the method.

---

> ### Author Response · Authors · 2021-08-26
> **Response to Reviewer QM46**
>
> We thank the reviewer for constructive feedback.
>
> > Are the authors open to releasing code and data upon a potential acceptance?
>
> Our code/data/models (including code for training the models) are prepped and ready for release. We plan on releasing it when the paper is accepted for publication.
>
> > How are 'training' samples constructed? Do they take into account potentially wrong initial segments of baseline detectors?
>
> Thank you for pointing out the lack in detail. As mentioned in L48-L51 in the Supplement, we train SGS-Net by randomly splitting/merging/adding/deleting masks in the same fashion as detailed in Xie et al. [7]. In more detail, we take an initial segmentation (this can be predicted from something such as UOIS-Net from Xie et al. [7], or it could be the ground truth label), perturb it with random splitting/merging/adding/deleting as per [7], then compute the ground truth score .8F + .2F@.75 (L191 in the main paper) of this perturbed segmentation w.r.t. the ground truth label. SGS-Net is trained to regress to this score.
>
> For SplitNet, given an initial segmentation, we randomly choose neighboring masks to merge, and SplitNet is trained with BCE loss to predict the ground truth split boundary.
>
> For DeleteNet, given an initial segmentation, we randomly add masks in the same fashion as Xie et al. [7], and train DeleteNet with BCE loss to predict binary score of deleting the falsely added masks. We will expand these details in a revised supplement.
>
> > How many of the baselines use the same input modalities (RGB-D) as the presented method? How many use just RGB or D? Might some of the gains come from these additional modalities?
>
> All baselines use RGB-D as input. RICE, which also consumes RGB-D, does not gain any performance due to additionally sensed modalities. We will clarify this in a revision.
>
> > Could RICE work from scratch? Why does it need a baseline detector? Even with the additional details in the supplement, it would be hard to reproduce without code, which greatly diminishes the value of the submission.
>
> As it is structured, RICE requires an initial segmentation in order to work. While we have fed it strong baseline detectors (Mask R-CNN, PointGroup, UOIS-Net, and UCN), it is feasible to supply it with the output of a traditional segmentation method such as GraphCuts. As mentioned, our code/data/models will be made publicly available upon publication in order to improve the reproducibility of our work.
>
> > SplitNet could as well just predict a binary mask instead of its boundary that needs to be sampled and optimized... It would be nice to know that the presented boundary prediction is in deed somewhat optimal.
>
> The main reason why the SplitNet output of boundary probabilities (that need to be sampled and optimized) is better than the binary mask is that a predicted binary mask may result in non-connected components. A binary mask can result in an arbitrary topology, which can be confusing for splitting a single mask into two connected components. Instead, our method provides a sampled path through the mask which will split the mask into two pieces. As the reviewer (QM46) mentioned, there is also ambiguity with predicting mask 1 vs mask 2 (e.g. setting the left mask to be mask 1 introduces a bias in the labels), which is not the case when predicting the split boundary. Lastly, sampling from the predicted split boundary adds more uncertainty to RICE, and we show this is useful in identifying and understanding cluttered scenes.
>
> > In a similar spirit, the AddNet relies on a proposal constructed from a connected component, why not use an RPN?
>
> While we could have ablated this design choice, the main focus of this paper is the introduction of SO-Nets, SGS-Nets, and the entire framework (RICE). Thus, we decided to focus our experiments on understanding the framework and why it works. Additionally, adding was not as crucial as splitting, so we chose a simple method of adding masks and did not focus our efforts on this less important design choice.
>
> > In addition, it would be nice to have a more detailed comparison of SplitNet and the entire pipeline, as the performance if split alone is almost the same as the complete combinatorial optimization.
>
> In the Section 6.2 of the Supplement, we provide a detailed analysis of RICE when only using 1 type of sampling operation (e.g. split only, merge only, add/delete only). Table 1 in the Supplement indeed shows that the split operation is very important to achieving the performance of the entire framework of RICE. This is in part because the initial segmentation methods (Mask R-CNN, PointGroup, UOIS-Net, UCN) tend to segment the objects together, which is a typical failure method in heavy clutter. Missing detections is relatively rare. They thereby benefit more from splits. Split only can be compared with Merge only, and the Supplement text (L86 - L103) discusses this experiment.

---

### Meta-Review · Area_Chair_eDPg · 2021-08-13

**Recommendation:** Accept (Poster)
**Confidence:** 4

**Metareview:**

All authors agree that the proposed method is interesting, novel and applicable to a wide array of segmentation algorithms. All reviewers agree that the paper is well-written and criticism relates to its reproducibility and design choices (QM46) or choices in the evaluation (5yWX, cMBw). However, three of four reviewers recommend acceptance (1x strong accept, 2x weak accept, 1x weak reject). Although this paper may be a better fit for a computer vision conference, I am following this assessment.

---

> ### Author Response · Authors · 2021-08-26
> **Response to Meta Review**
>
> We thank the meta-reviewer for the helpful feedback.
>
> > criticism relates to its reproducibility
>
> Our code/data/models (including code for training the models) are prepped and ready for release. We plan on releasing the code when the paper is accepted for publication.
>
> > this paper may be a better fit for a computer vision conference
>
> As we mention in our response to reviewer cMBw, our method is directly motivated by robot requirements, and exploits the nuances of cluttered manipulable object settings, which are common to robot manipulation environments, and not necessarily in-the-wild computer vision settings.

---

### Decision · Program_Chairs · 2021-09-13

**Decision:**

Accept (Poster)

**Comment:**

All authors agree that the proposed method is interesting, novel and applicable to a wide array of segmentation algorithms. All reviewers agree that the paper is well-written and criticism relates to its reproducibility and design choices (QM46) or choices in the evaluation (5yWX, cMBw). However, three of four reviewers recommend acceptance (1x strong accept, 2x weak accept, 1x weak reject). Although this paper may be a better fit for a computer vision conference, I am following this assessment.